# The Stability Analysis of a Tether for a Segmented Space Elevator

Shihao Luo [1] , Naigang Cui [2], Xiaowei Wang [3], Youhua Fan [1],* and Haitao Gu [1]

[1] School of Science, Harbin Institute of Technology (Shenzhen), Shenzhen 518055, China; luoshihao@stu.hit.edu.cn (S.L.); 16s057430haitao@alu.hit.edu.cn (H.G.)
[2] School of Astronautics, Harbin Institute of Technology, Harbin 150001, China; cuinaigang1965@hit.edu.cn
[3] R&D Center, China Academy of Launch Vehicle Technology, Beijing 100076, China; wangxwbuaa@163.com
* Correspondence: yhfan@hit.edu.cn

**Abstract:** The space elevator system is a space tether system used to solve low-cost space transportation. Its high efficiency, large load and other characteristics have broad application prospects in the aerospace field. The stability analysis is the foundation of the space elevator system research. Based on the new segment space elevator system model, in this paper, the stability of the system at the equilibrium point is analyzed by Lyapunov stability theory; And based on the criterion that the change rate of the system restoring torque and the anchor point tension are greater than 0, the maximum offset angle of the system inside and outside the equatorial plane is analyzed. The results show that the segment space elevator is stable near the equilibrium point; The maximum deflection angle of the space elevator inside and outside the equatorial plane is related to the design stress of the anchor point; When the space elevator is offset outside the equatorial plane, it will only lose stability because the restoring torque reaches the maximum value; When the space elevator is offset in the equatorial plane, and due to the design stress of the anchor point is small, it will lose stability because the tensile force of the anchor point is reduced to 0, and when the design stress of the anchor point is large, it will lose stability because the recovery torque reaches the maximum value; The stability of the space elevator outside the equatorial plane is better than that in the equatorial plane.

**Keywords:** space elevator; stability; segment





## 1. Introduction

As early as 1895, Konstantin Tsiolkovsky, the father of aerospace, put forward the concept of space elevator (SE): build a high tower from the equator and connect the ground with the space station in geostationary orbit (GEO). When the design parameters are appropriate, the gravity received by the space elevator and the centrifugal force rotating with the earth offset, and the resultant force received by the space elevator on the ground can be zero.

The advantage of the space elevator is that it can continuously transport goods to the space station. It is estimated that hundreds of tons of goods can be transported to the space station every week [1]. At present, the carrying capacity of the "heavy Falcon" carrier rocket with the largest thrust in the world to reach GEO at a single time is no more than 26.7 tons, and it requires a long launch preparation time. In addition, the transportation cost of the space elevator also has great advantages over the launch vehicle. At present, the estimated transportation cost of space elevator can be less than $100/kg, while the carrier rocket is much higher. For example, the cost of transporting the "heavy Falcon" carrier rocket to GEO reaches $3370/kg, and space elevator may use solar energy to provide energy in the future [2]. Therefore, space elevator is superior to the traditional carrier rocket in economy and environmental protection.

Although the space elevator has many advantages, in 1979, Pearson found that the strength of the materials used to manufacture the space elevator was much higher than that

of conventional materials such as steel, which affected the feasibility of the space elevator system [3]. Until 1991, people discovered carbon nanotube materials, which made the research of space elevator return to the dawn. The tensile strength of carbon nanotubes can reach more than 100 times that of steel, which can meet the strength required for the manufacture of space elevator. Moreover, China's scientific research team can manufacture half meter long carbon nanotubes in 2013 [4]. In the future, people will be able to manufacture larger carbon nanotubes, making the manufacture of space elevator possible.

For the design of the space elevator system, the biggest difficulty lies in the excessive axial force on the rope when it reaches the force balance. Through estimation, Aravind found that when the section of the space elevator does not change with the height, that is, under the condition of constant section space elevator, the normal stress of the rope reaches 382 GPa, which is much higher than the tensile strength of traditional materials such as steel [5]. Although the ideal strength of carbon nanotubes is expected to meet the demand [6], reducing the working stress of the rope of the space elevator system is still the most important problem in the design of the space elevator. The section of the space elevator is set to change with the height, so that the stress does not change with the height, that is, the variable section space elevator can effectively reduce the stress of the rope. Cohen and Misra gave the expression of cross-sectional area of variable cross-section space elevator under equal stress state in 2007 [7]. In this case, the stress of space elevator can be significantly reduced. At present, most researches on the space elevator are based on the space elevator with variable section. Inspired by biology, Dan and sun proposed that the safety margin of the space elevator structure can be reduced, that is, the working stress ratio can be improved in the space elevator design, and the material can repair itself at any time. Through calculation, it is found that in this case, the space elevator made of M5 material can meet the design requirements [8]. Shi and Luo put forward the concept of segmented space elevator, that is, the sectional area of the space elevator is changed in sections, and made a preliminary study on its mechanical problems [9,10]. This segmented space elevator has less stress than the constant section space elevator and is easier to process and manufacture than the variable section space elevator. It can be built in sections up and down at the same time from the space station on the synchronous orbit. Compared with the previous design schemes, it has many advantages and provides a new idea for the future space elevator design.

There are other space elevator model designs, such as the non-equatorial space elevator system. The ground anchor point of the equatorial space elevator can only be selected on the equator, which limits the construction site of the space elevator system. The earth's non-equatorial space elevator built in the low latitude area of non-equatorial can solve this problem, but in this case, the gravity of the rope is not in the axial direction, which increases the difficulty of its design. Gassend gave an approximate solution of the cross-sectional area of the earth's non-equatorial space elevator [11]. Wang Established the static model of the earth's non-equatorial space elevator in 2019, found that increasing the tensile strength of the tether material can expand the deployment range of the space elevator system [12], established the dynamic equation and analyzed its vibration mode [13]. Okino proposed a new type of counterweight space elevator system. The system is similar to the ground elevator and consists of two cables: one guide cable bears the tension applied to the structure, that is, the rope of the conventional elevator, and the other moving cable connects the two gondolas to move up and down respectively. The performance of counterweight space elevator is analyzed by numerical calculation [14]. Li proposed a new concept of multi climbing rope ring tether transportation system (L-TTS) for efficient transportation of payload. It consists of two parallel tether transportation systems or part of the space elevator, and each tether has multiple climbers. It will reduce the overall vibration of the system, but there is a risk of tether collision during load transportation [15]. On this basis, the team proposed a new type of annular rope carrier transportation system (L-TTS-R). In this new concept, in addition to the components mentioned in the L-TTS, the system also includes several parallel rigid rings, which are evenly fixed on two tethers to keep the

distance between the two connection points unchanged. The effects of steps on system vibration suppression, climber collision risk avoidance, platform relative oscillation and pitch motion and tether tension are evaluated [16].

The scale of the space elevator rope is large, and the operating environment is complex. The vibration caused by various factors may have a serious impact on the operation of the space elevator system and endanger the reliability and safety of the space elevator system. Therefore, the dynamics of space elevator system has attracted extensive attention in recent years. Williams studied the rope vibration characteristics caused by the movement of the climber based on the bead model, established the dynamic and kinematic models in the rope vibration process respectively, and carried out the modal analysis on this basis [17]. The dynamics of the climbing rope system can be described by the Euler-Lagrange method, and the high-frequency tether can be used to capture the position of the flexible rope system [18]. Hu studied the vibration problem of super flexible damping space structure, considered the coupling between structural vibration, attitude dynamics and track dynamics, and studied the vibration characteristics and wave propagation characteristics of space flexible damping plates with four special springs [19]. Yoon established a proportional experimental model considering the initial tension of tether in 2020, studied the band gap characteristics of meta-materials through experiments, and measured the deformation shape of meta-materials in the band gap [20]. Luo analyzed the dynamics of the space elevator system based on the absolute node coordinate formula (ANCF), and found that this method can achieve the same calculation accuracy with fewer elements, and has a faster convergence speed [21].

Previous studies have focused on the external excitation of the space elevator system, such as the Coriolis force caused by the climber, the oscillation period and response of the rope when impacted. The stability of the space elevator system itself has not been analyzed and studied in detail. Compared with the previous design scheme, the segment space elevator model has many advantages and is a possibility for the construction scheme of the space elevator system in the future. Therefore, based on the model of segment space elevator system, the Lyapunov stability theory is applied to study the stability of space elevator system at the equilibrium point of the system. Then, by analyzing the maximum deflection angle of the space elevator system inside and outside the equatorial plane, the stability range of the space elevator system near the equilibrium point is studied. This work provides a basis for further research of segment space elevator system, such as the oscillation suppression in the future.

## 2. Stability of Equilibrium Point

According to the design in literature [10], the design of the segmented space elevator system model is to subdivide the rope into several segments along the length direction. Each segment of rope can be composed of several equal cross-section tethers in different numbers and connected by a connecting platform. The model is shown in Figure 1.

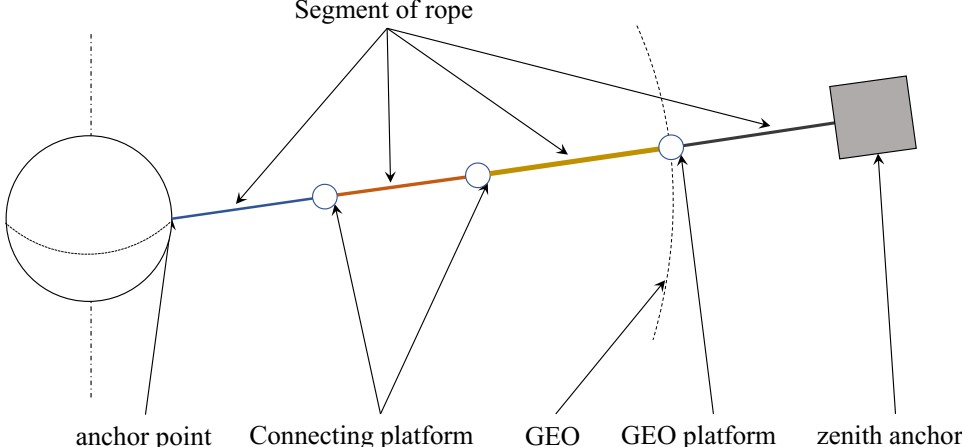

**Figure 1.** Schematic diagram of segmented space elevator system model reprint from [10].

For a two-segments space elevator system, a certain amount of lateral force is applied to the direction outside the equatorial plane at the position of the zenith anchor respectively. The results calculated by using the ANCF method in [21] are shown in Table 1 and Figure 2.

**Table 1.** Calculation results of lateral loading at zenith anchor.

| Lateral Force | Zenith Anchor Displacement in Z Direction | Offset Angle |
|---|---|---|
| $1 \times 10^6$ N | $7.31 \times 10^4$ m | 0.001048 rad |
| $1 \times 10^7$ N | $7.31 \times 10^5$ m | 0.010479 rad |
| $5 \times 10^7$ N | $3.66 \times 10^6$ m | 0.052490 rad |
| $1 \times 10^8$ N | $7.31 \times 10^6$ m | 0.105577 rad |
| $2 \times 10^8$ N | $1.46 \times 10^7$ m | 0.216145 rad |
| $3 \times 10^8$ N | $2.19 \times 10^7$ m | 0.338005 rad |
| $5 \times 10^8$ N | $3.61 \times 10^7$ m | 0.665371 rad |

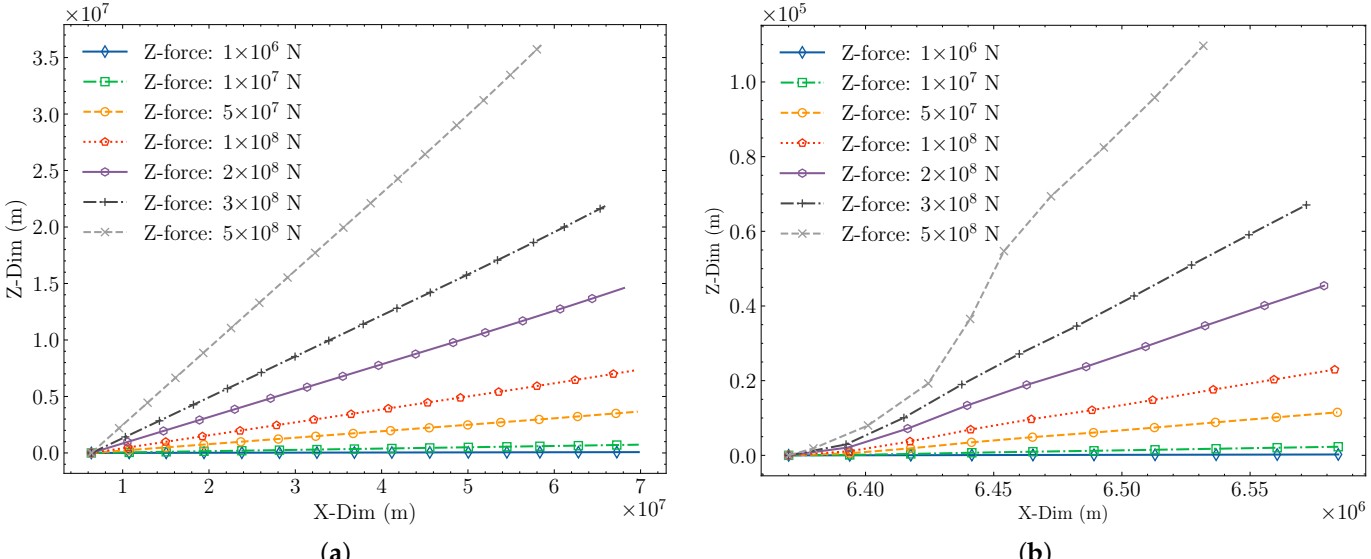

**Figure 2.** Calculation results of lateral loading at zenith anchor. (**a**) Deformation of the rope by lateral force loading. (**b**) Deformation of the rope near the anchor point on the ground by lateral force loading.

It can be seen from Figure 2b that when the rope of the space elevator system is laterally deformed, the stress at the anchor end of the ground is small, the main bending part is near the ground anchor point. And the bending of the rope increases as the deflection angle increases. From Figure 2a, the overall shape of the rope remains roughly straight, and is independent of the deflection angle. Therefore, the rope of the space elevator system can be simplified as a rigid rod, which is connected with the ground anchor point with a spherical hinge. The schematic diagram of the model is shown in Figure 3.

The center of the Earth $O$ is assumed to be fixed and is used as the origin of the inertial frame. $O_0$ is the anchor point of the tether on the earth's equator. The $X$ axis points from point $O$ to point $O_0$ and is perpendicular to the ground. The $Z$ axis points to the North Pole along the rotation axis of the Earth. The $Y$ axis is perpendicular to the $X$ axis and $Z$ axis, $\omega$ is the angular velocity of the system equal to the rotation rate of the Earth. $l$ is the nominal (unstressed) length of the tether. $m_c$ is the mass of the counterweight at the end of the tether. $\theta$ is the angle between the projection of the rope of the space elevator system in the $XY$ plane and the $X$ axis. $\phi$ is the angle between the projection of the rope of the space elevator system in the $XY$ plane and itself.

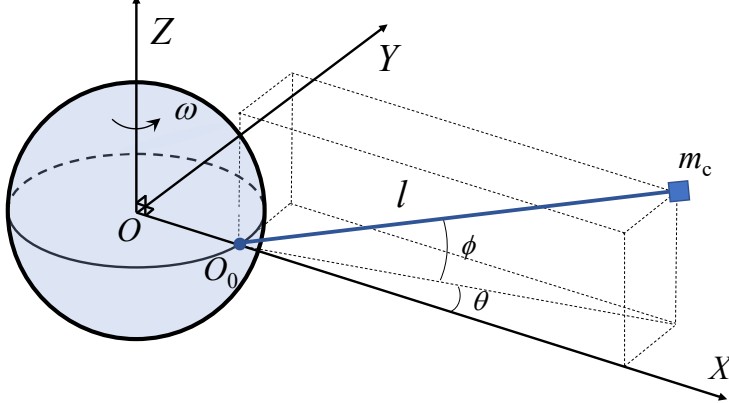

**Figure 3.** Schematic diagram of rigid rope model.

The position vectors of zenith anchor and rope element in the *O-XYZ* coordinate system are $\mathbf{r}_c$ and $\mathbf{r}_s$, respectively, where *s* is the position of the rope element on the tether. The position vectors of each part of the system can be expressed as

$$\mathbf{r}_c = (R_e + l \cos\theta \cos\phi)\mathbf{i} + (l \sin\theta \cos\phi)\mathbf{j} + (l \sin\phi)\mathbf{k}, \tag{1}$$

$$\mathbf{r}_s = (R_e + s \cos\theta \cos\phi)\mathbf{i} + (s \sin\theta \cos\phi)\mathbf{j} + (s \sin\phi)\mathbf{k}, \tag{2}$$

where $R_e$ is the radius of the Earth.

By differentiating the time through Equations (1) and (2), the velocity vectors of rope element and zenith anchor in Cartesian coordinate system can be obtained as $\dot{\mathbf{r}}_s$ and $\dot{\mathbf{r}}_c$ respectively. The total kinetic energy of the system consists of the kinetic energy of the zenith anchor and the kinetic energy of the rope from the equatorial ground anchor to the zenith anchor, which can be expressed as in Equation (3):

$$T = \frac{1}{2}m_c\dot{\mathbf{r}}_c^\top \dot{\mathbf{r}}_c + \frac{1}{2}\int_{R_e}^{l} \rho A(r)\dot{\mathbf{r}}_s^\top \dot{\mathbf{r}}_s \mathrm{d}r, \tag{3}$$

where $\rho$ is the density of the rope material, $A(r)$ is the cross-sectional area at the position *s* and *r* is the coordinate of the rope element of the space elevator in the geocentric coordinate system, which can be expressed as

$$r = R_e + s. \tag{4}$$

The total potential energy *V* of the system is expressed in Equation (5):

$$V = -\frac{\mu m_c}{\sqrt{\mathbf{r}_c^\top \mathbf{r}_c}} - \int_{R_e}^{l} \frac{\mu \rho A(r)}{\sqrt{\mathbf{r}_s^\top \mathbf{r}_s}} \mathrm{d}r, \tag{5}$$

where $\mu$ is the gravitational constant of the earth.

Due to the internal friction in the structure, the influence of atmospheric resistance near the ground and other factors, the force conditions inside and outside the equatorial plane of the space elevator system are different, the damping coefficients inside and outside the plane are $c_1$ and $c_2$ respectively. The work dissipated is expressed in Equation (6):

$$\Phi = \int_0^{\dot{\theta}} c_1\dot{\theta}\mathrm{d}\dot{\theta} + \int_0^{\dot{\phi}} c_2\dot{\phi}\mathrm{d}\dot{\phi}$$
$$= \frac{1}{2}c_1\dot{\theta}^2 + \frac{1}{2}c_2\dot{\phi}^2. \tag{6}$$

The Lagrange equation is shown below:

$$\frac{\mathrm{d}}{\mathrm{d}t}\left(\frac{\partial(T-V)}{\partial\dot{q}_i}\right) - \frac{\partial(T-V)}{\partial q_i} + \frac{\partial\Phi}{\partial\dot{q}_i} = 0. \tag{7}$$

By substituting Equations (3)–(6) into Equation (7), the dynamics equation of $\theta$ and $\phi$ of the system can be obtained:

$$\ddot{\theta} = -\frac{c_1}{M_1 \cos^2 \phi}\dot{\theta} + 2\dot{\phi}(\dot{\theta} + \omega)\tan\phi$$
$$-\frac{M_2 R_e \omega^2 \sin\theta}{M_1 \cos\phi} + \frac{\mu R_e M_3 \sin\theta}{M_1 \cos\phi}, \tag{8a}$$

$$\ddot{\phi} = -\frac{c_2}{M_1}\dot{\phi} - (\dot{\theta} + \omega)^2 \cos\phi \sin\phi$$
$$-\frac{M_2}{M_1}\omega^2 R_e \cos\theta \sin\phi \tag{8b}$$
$$+\frac{M_3}{M_1}\mu R_e \cos\theta \sin\phi,$$

where $\omega$ is the angular velocity of the earth's rotation and

$$M_1 = m_c l^2 + \int_{R_e}^{l} \rho A(r) r^2 \mathrm{d}r, \tag{9a}$$

$$M_2 = m_c l + \int_{R_e}^{l} \rho A(r) r \mathrm{d}r, \tag{9b}$$

$$M_3 = \frac{m_c l}{(\mathbf{r}_c^\top \mathbf{r}_c)^{\frac{3}{2}}} + \int_{R_e}^{l} \frac{\rho A(r) r}{(\mathbf{r}_s^\top \mathbf{r}_s)^{\frac{3}{2}}}\mathrm{d}r. \tag{9c}$$

According to Equations (8) and (9), the dynamics equation of the system is a second-order nonlinear system with variable damping and stiffness.

Based on the small-angle hypothesis: $\cos\theta \approx 1$, $\cos\phi \approx 1$, $\sin\theta \approx \theta$ and $\sin\phi \approx \phi$, Equations (8) and (9) can be simplified as

$$\ddot{\theta} = -\frac{c_1}{M_1}\dot{\theta} + 2\dot{\phi}(\dot{\theta} + \omega)\phi - \frac{M_2 R_e \omega^2 \theta}{M_1} + \frac{\mu M_3 R_e \theta}{M_1}, \tag{10a}$$

$$\ddot{\phi} = -\frac{c_2}{M_1}\dot{\phi} - (\dot{\theta} + \omega)^2 \phi - \frac{M_2}{M_1}\omega^2 R_e \phi + \frac{\mu M_3 R_e \phi}{M_1}. \tag{10b}$$

After Equation (10) is sorted out, the matrix form of the dynamics equation of the system is shown:

$$\begin{bmatrix} \ddot{\theta} \\ \ddot{\phi} \end{bmatrix} + \mathbf{C}\begin{bmatrix} \dot{\theta} \\ \dot{\phi} \end{bmatrix} + \mathbf{K}\begin{bmatrix} \theta \\ \phi \end{bmatrix} = \mathbf{\Psi}, \tag{11}$$

where $\mathbf{C}$ is the damping of the system, $\mathbf{K}$ is the stiffness of the system, and $\mathbf{\Psi}$ is the non-linearity of the system, which are respectively expressed in Equation (12):

$$\mathbf{C} = \begin{bmatrix} c_1/M_1 & 0 \\ 0 & c_2/M_1 \end{bmatrix}, \tag{12a}$$

$$\mathbf{K} = \begin{bmatrix} J_1 & 0 \\ 0 & J_2 \end{bmatrix}, \tag{12b}$$

$$\mathbf{\Psi} = \begin{bmatrix} 2\dot{\phi}(\dot{\theta} + \omega)\phi \\ -\dot{\theta}(\dot{\theta} + 2\omega)\phi \end{bmatrix}, \tag{12c}$$

where

$$J_1 = \frac{M_2 \omega^2 - M_3 \mu}{M_1}R_e, \tag{13a}$$

$$J_2 = \frac{M_2 \omega^2 - M_3 \mu}{M_1}R_e + \omega^2. \tag{13b}$$

Let $x_1 = \theta$, $x_2 = \dot{\theta}$, $x_3 = \phi$ and $x_4 = \dot{\phi}$, Equation (11) can be converted into four first-order differential equations:

$$
\begin{cases}
\dot{x}_1 = x_2 \\
\dot{x}_2 = -\dfrac{c_1}{M_1} x_2 - J_1 x_1 + 2x_4(x_2 + \omega)x_3 \\
\dot{x}_3 = x_4 \\
\dot{x}_4 = -\dfrac{c_2}{M_1} x_4 - J_2 x_3 - x_2(x_2 + 2\omega)x_3.
\end{cases}
\tag{14}
$$

The system dynamics equation is as follows:

$$
\begin{bmatrix} \dot{x}_1 & \dot{x}_2 & \dot{x}_3 & \dot{x}_4 \end{bmatrix}^{\mathsf{T}} = \mathbf{0}.
\tag{15}
$$

The equilibrium point of Equation (15) can be easily found:

$$
\mathbf{x}_0 = \begin{bmatrix} 0 & 0 & 0 & 0 \end{bmatrix}^{\mathsf{T}}.
\tag{16}
$$

According to the above formula, it can be found that under the action of earth gravity, the equilibrium point of the system is located at zero point. According to Equation (14), it can be found that there are nonlinear terms in the system dynamics equation, so the above dynamics equation is a coupling nonlinear dynamics equation. In order to make it easy for us to judge the stability of the system, we need to linearize the nonlinear system first, and the main idea of linearization is first-order approximation. Taylor's expansion method is used to linearize nonlinear equations near singularities.

Equation (14) of the nonlinear equations is rewritten into matrix form, and the perturbation solution $\mathbf{x}(t) = \mathbf{x}_0(t) + \delta\mathbf{x}(t)$ near the equilibrium point is taken. After omitting the higher-order term, the following equation can be obtained:

$$
\delta\dot{\mathbf{x}}(t) = \mathbf{A}\delta\mathbf{x}(t),
\tag{17}
$$

where $\mathbf{A}$ is the Jacobian matrix of the system, which is shown as:

$$
\mathbf{A} = \begin{bmatrix}
0 & 1 & 0 & 0 \\
-J_1 & -c_1/M_1 & 0 & 0 \\
0 & 0 & 0 & 1 \\
0 & 0 & -J_2 & -c_2/M_1
\end{bmatrix}.
\tag{18}
$$

The eigenvalue of the matrix $\mathbf{A}$ can be solved by Equation (19):

$$
\det(\lambda\mathbf{E} - \mathbf{A}) = \mathbf{0}.
\tag{19}
$$

And the eigenvalue of $\mathbf{A}$ can be obtained:

$$
\begin{cases}
\lambda_{1,2} = \dfrac{-c_1 \pm \sqrt{\Delta_1}}{2M_1} \\
\lambda_{3,4} = \dfrac{-c_2 \pm \sqrt{\Delta_2}}{2M_1}.
\end{cases}
\tag{20}
$$

where

$$
\begin{aligned}
\Delta_1 &= c_1^2 - 4J_1 M_1^2 \\
\Delta_2 &= c_2^2 - 4J_2 M_1^2
\end{aligned}
\tag{21}
$$

According to Lyapunov stability theory, the stability of the system near the equilibrium point is judged by the positive and negative of discriminant $\Delta_1$ and $\Delta_2$ of eigenvalues in Equation (20).

### 3. Stability Range near the Equilibrium Point

The stability range of the system near the equilibrium point is also a very important problem. There are two reasons for the instability of the system caused by the lateral displacement of the space elevator system:

First, the lateral displacement of the space elevator system will cause the direction of universal gravitation and centrifugal force to no longer follow the direction of the rope, but will produce the torque to rotate the whole space elevator system. When the torque to restore the space elevator system $M_r$ to a stable position is reduced, the space elevator system will become unstable;

Second, the lateral displacement of the space elevator system will change the magnitude of the universal gravitation and centrifugal force of the system, which will lead to the change of the tensile force of the ground anchor point with the least stress. When the tensile force of the ground anchor point $T_0$ is reduced to 0, the space elevator system will become unstable.

Since the centrifugal force direction of the system is always perpendicular to the rotation axis of the earth, the force of the system offset in and out of the equatorial plane is different, which needs to be analyzed separately.

#### 3.1. Offset Outside the Equatorial Plane of the Earth

As shown in Figure 4, $\phi$ is the deflection angle of the system outside the equatorial plane, $s$ is the coordinate of a micro element on the system rope in the length direction of the rope, $r$ is the distance from the rope element to the center of the earth, $F_g$ is the universal gravitation of the micro element by the earth, and $F_c$ is the centrifugal force of the micro element.

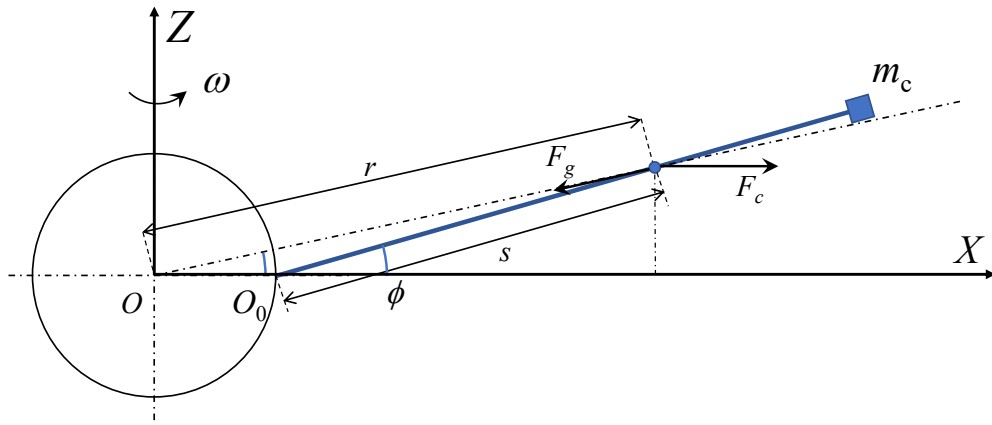

**Figure 4.** Analysis diagram of offset outside equatorial plane.

Assuming that the anchor point $O_0$ of the rope is hinged, the rope is rigid, and the counterclockwise torque is positive. The distance from any point on the rope to the center of the earth can be expressed as

$$r_o = \sqrt{(s \sin \phi)^2 + (R_e + s \cos \phi)^2}. \tag{22}$$

The moments of gravity and centrifugal force on the rope element at the position $r_o$ to the ground anchor point are $dM_{go}$ and $dM_{co}$ respectively, which can be expressed as

$$dM_{go} = \frac{\mu A(s)\rho ds}{r_o^2} * \frac{R_e s \sin \phi}{r_o}, \tag{23a}$$

$$dM_{co} = \frac{\mu A(s)\rho ds(R_e + s \cos \phi)}{R_g^3} * s \sin \phi, \tag{23b}$$

where $R_g$ is the radius of the geosynchronous orbit.

In order to simplify the calculation, it is assumed that the segmented space elevator system has 2 segments, and the cross-sectional area of each segment is designed according to the segment function of Equation (24).

$$A(s) = \begin{cases} a_0, & 0 <= s <= p_s * R_e; \\ p_a * a_0, & p_s * R_e < s <= p_l * R_e, \end{cases} \tag{24}$$

where $a_0$ is the initial cross-sectional area of the rope, $p_a$ is the growth rate of rope cross-sectional area, $p_s$ is the ratio of the coordinate of the segment point to the length of the earth's radius and $p_l$ is the ratio of the total length of the space elevator system to the length of the earth's radius.

The torque of the zenith anchor is expressed as

$$M_{tgo} = \frac{\mu p_l m_c R_e^2}{r_{lo}^3} \sin \phi, \tag{25a}$$

$$M_{tco} = \frac{\mu p_l m_c R_e^2}{R_g^3} \sin \phi (p_l \cos \phi + 1), \tag{25b}$$

where

$$r_{lo} = R_e \sqrt{p_l^2 + 2p_l \cos \phi + 1}. \tag{26}$$

The restoring torque can be expressed as

$$\begin{aligned} M_{ro} &= M_{go} + M_{tgo} - M_{co} - M_{tco} \\ &= \int_0^{p_l * R_e} dM_{go} + M_{tgo} - \int_0^{p_l * R_e} dM_{co} - M_{tco}. \end{aligned} \tag{27}$$

For the force in $X$ direction, it can be deduced that the force of rope micro element are

$$dF_{go} = \frac{\mu A(s)\rho ds}{r_o^2} * \frac{R_e + s \cos \phi}{r_o}, \tag{28a}$$

$$dF_{co} = \frac{\mu A(s)\rho ds (R_e + s \cos \phi)}{R_g^3}. \tag{28b}$$

The force of the zenith anchor is expressed as

$$F_{tgo} = \frac{\mu m_c R_e}{r_{lo}^3} (p_l \cos \phi + 1), \tag{29a}$$

$$F_{tco} = \frac{\mu m_c R_e}{R_g^3} (p_l \cos \phi + 1). \tag{29b}$$

The force in the $X$ direction of the anchor point can be expressed as

$$\begin{aligned} T_o &= \int_0^{p_l * R_e} dF_{co} + F_{tco} - \int_0^{p_l * R_e} dF_{go} - F_{tgo} \\ &= F_{co} + F_{tco} - F_{go} - F_{tgo}. \end{aligned} \tag{30}$$

The detailed calculation formula can be found in Appendix A.1. Through the restoring torque of the system and the tension of the anchor point, the stability range of the space elevator system when it is offset outside the equatorial plane can be obtained.

### 3.2. Offset in the Equatorial Plane of the Earth

The stress of the space elevator system when it is offset in the equatorial plane is shown in Figure 5, $\theta$ is the deflection angle of the system outside the equatorial plane and other parameters are the same as those outside the equatorial plane.

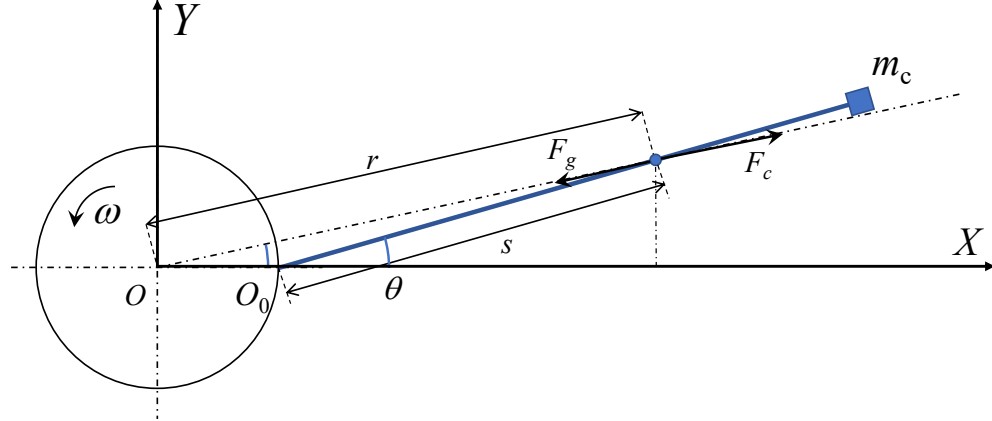

**Figure 5.** Analysis diagram of offset in equatorial plane.

It can be found that when the system is offset in the equatorial plane, the only change is the centrifugal force, and the distance from any point on the rope to the center of the earth can be expressed as

$$r_i = \sqrt{(s \sin \theta)^2 + (R_e + s \cos \theta)^2}. \tag{31}$$

The torque $dM_{gi}$ and $dM_{ci}$ of the rope element are expressed as

$$dM_{gi} = \frac{\mu A(s)\rho ds}{r_i^2} * \frac{R_e s \sin \theta}{r_i}, \tag{32a}$$

$$dM_{ci} = \frac{\mu A(s)\rho ds(R_e + s \cos \theta)}{R_g^3} * \frac{R_e s \sin \theta}{r_i}. \tag{32b}$$

The torque of the zenith anchor is expressed as:

$$M_{tgi} = \frac{\mu m_c}{r_{li}^2} * \frac{R_e l \sin \theta}{r_{li}}, \tag{33a}$$

$$M_{tci} = \mu m_c \frac{(R_e + l \cos \theta)}{R_g^3} * \frac{R_e l \sin \theta}{r_{li}}, \tag{33b}$$

where

$$r_{li} = R_e \sqrt{p_l^2 + 2p_l \cos \theta + 1}. \tag{34}$$

The restoring torque can be expressed as

$$\begin{aligned} M_{ri} &= M_{gi} + M_{tgi} - M_{ci} - M_{tci} \\ &= \int_0^{p_l * R_e} dM_{gi} + M_{tgi} - \int_0^{p_l * R_e} dM_{ci} - M_{tci}. \end{aligned} \tag{35}$$

For the force in $X$ direction, it can be deduced that the force of rope micro element are

$$\mathrm{d}F_{gi} = \frac{\mu A(s)\rho \mathrm{d}s}{r_i^2} * \frac{R_e + s\cos\theta}{r_i},$$

(36a)

$$\mathrm{d}F_{ci} = \frac{\mu A(s)\rho \mathrm{d}s(R_e + s\cos\theta)}{R_g^3} * \frac{R_e + s\cos\theta}{r_i}.$$

(36b)

The force of the zenith anchor is expressed as

$$F_{tgi} = \frac{\mu m_c R_e}{r_{li}^3}(p_l\cos\theta + 1),$$

(37a)

$$F_{tci} = \frac{\mu m_c R_e^2}{R_g^3 r_{li}}(p_l\cos\theta + 1)^2.$$

(37b)

The force in the $X$ direction of the anchor point can be expressed as

$$T_i = \int_0^{p_l * R_e} \mathrm{d}F_{ci} + F_{tci} - \int_0^{p_l * R_e} \mathrm{d}F_{gi} - F_{tgi}$$

$$= F_{ci} + F_{tci} - F_{gi} - F_{tgi}.$$

(38)

The detailed calculation formula can be found in Appendix A.2. Through the restoring torque of the system and the tension of the anchor point, the stability range of the space elevator system when it is offset outside the equatorial plane can be obtained.

## 4. Results

### 4.1. Stability of Equilibrium Point

The segment parameters of a 4-segments space elevator system are shown in Table 2 and the structural parameters are shown in Table 3.

**Table 2.** Segment parameters of the segment space elevator system.

| Segments | Segment Length | Cross-Sectional Area |
|---|---|---|
| Segment 1 | $10.779 \times 10^3$ km | $1.000 \times 10^{-2}$ m$^2$ |
| Segment 2 | $7.186 \times 10^3$ km | $1.123 \times 10^{-2}$ m$^2$ |
| Segment 3 | $17.965 \times 10^3$ km | $1.190 \times 10^{-2}$ m$^2$ |
| Segment 4 | $24.070 \times 10^3$ km | $2.000 \times 10^{-2}$ m$^2$ |

**Table 3.** System parameters of the segment space elevator system.

| Parameters | Value |
|---|---|
| Total length | $6.00 \times 10^4$ km |
| Rope density | $1.30 \times 10^3$ kg m$^{-3}$ |
| Mass of connecting platform | $1.00 \times 10^4$ kg |
| Mass of zenith anchor | $2.55 \times 10^9$ kg |

Through calculation, the stability analysis of the four segments space elevator system swinging in and out of the equatorial plane near the equilibrium point $x_0$ can be obtained as follows:

When $\Delta_1 > 0$ and $\Delta_2 > 0$, Equation (19) has four different negative real roots. It can be seen from Figure 6 that the nearby state vector fields all point to this equilibrium point and are finally stabilized at this equilibrium point. Therefore, the equilibrium point $x_0$ is a stable node, and the original nonlinear system is asymptotically stable near the equilibrium point $x_0$.

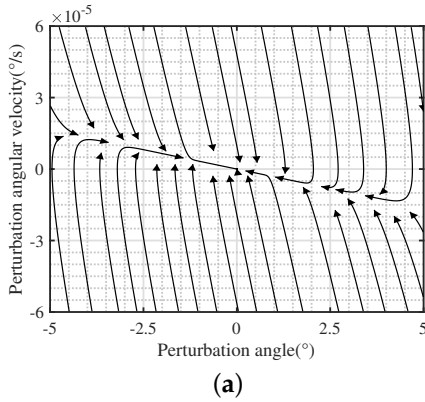 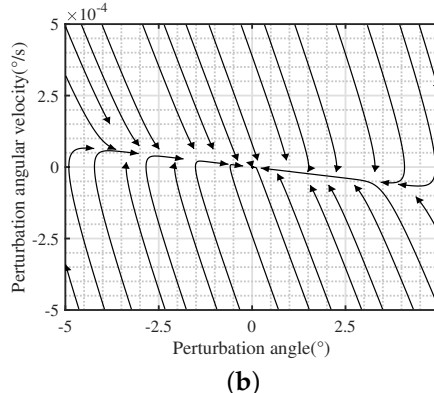

**(a)**  **(b)**

**Figure 6.** Phase diagram of system near $x_0$ ($\Delta_1 > 0$, $\Delta_2 > 0$). (**a**) In-plane swing $\theta$; (**b**) Out-plane swing $\phi$.

When $\Delta_1 < 0$ and $\Delta_2 < 0$, Equation (19) has two pairs of different conjugated virtual roots with negative real parts. It can also be seen from Figure 7 that the nearby state vector field is oriented to the equilibrium point and finally stabilized at the equilibrium point. Therefore, the equilibrium point $x_0$ is the stable focus, and the original nonlinear system is asymptotically stable near the equilibrium point $x_0$.

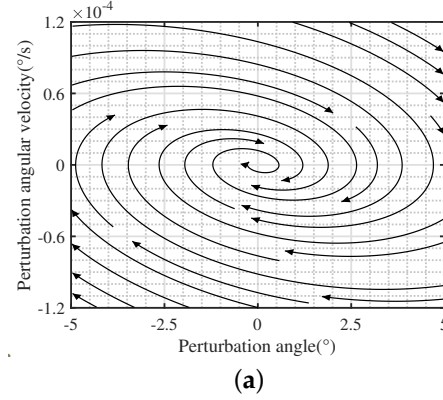 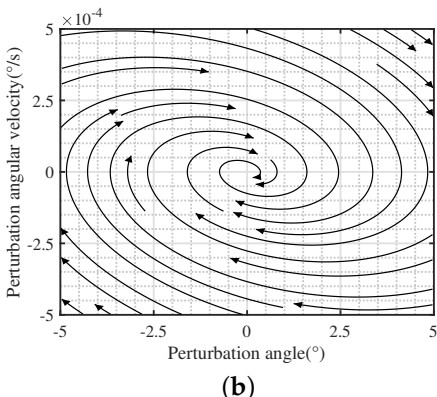

**(a)**  **(b)**

**Figure 7.** Phase diagram of system near $x_0$ ($\Delta_1 < 0$, $\Delta_2 < 0$). (**a**) In-plane swing $\theta$; (**b**) Out-plane swing $\phi$.

When $\Delta_1 = 0$ and $\Delta_2 = 0$, Equation (19) has four negative real roots that are equal in pairs. It can be seen from Figure 8 that the nearby state vector fields basically point to the equilibrium point in the same direction and finally stabilize at the equilibrium point. Therefore, the equilibrium point $x_0$ is a stable degenerate node, and the original nonlinear system is asymptotically stable near the equilibrium point $x_0$.

When $\Delta_1 > 0$, $\Delta_2 < 0$ or $\Delta_1 < 0$, $\Delta_2 > 0$, Equation (19) has a pair of negative real roots and a pair of conjugated imaginary roots with negative real parts. When the roots are negative real, the phase diagram are similar with Figure 6; when the roots are conjugated imaginary, the phase diagram are similar with Figure 7. The nearby state vector fields all point to this equilibrium point and eventually stabilize at this equilibrium point. Therefore, the linear system is strictly stable at $x_0$, and the original nonlinear system is asymptotically stable near the equilibrium point $x_0$.

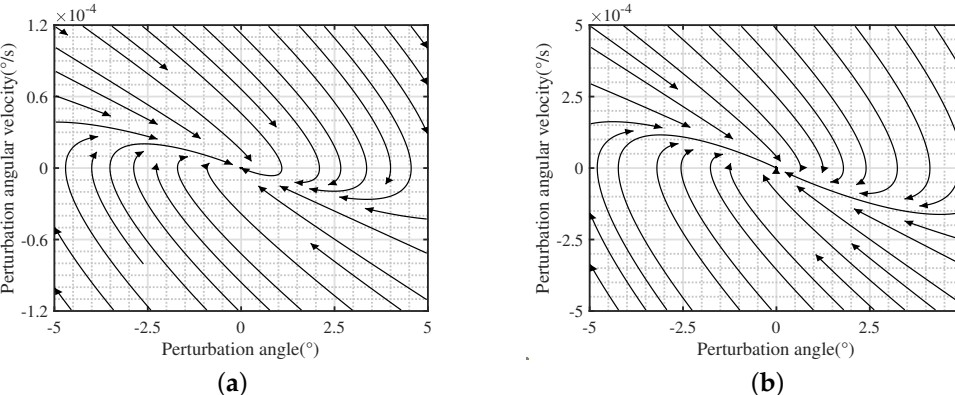

**Figure 8.** Phase diagram of system near $x_0$ ($\Delta_1 = 0$, $\Delta_2 = 0$). (**a**) In-plane swing $\theta$; (**b**) Out-plane swing $\phi$.

When $c_1 = 0$ and $c_2 = 0$, Equation (19) has two pairs of distinct conjugated pure imaginary roots. It can be seen from Figure 9 that the nearby state vector field is a circle of elliptic state vector fields, forming periodic orbits related to the initial state, and the system will eventually run in the orbit. Therefore, the equilibrium point $x_0$ is the center, and the system is a system without damp. The original nonlinear system will oscillate constant amplitude after being disturbed.

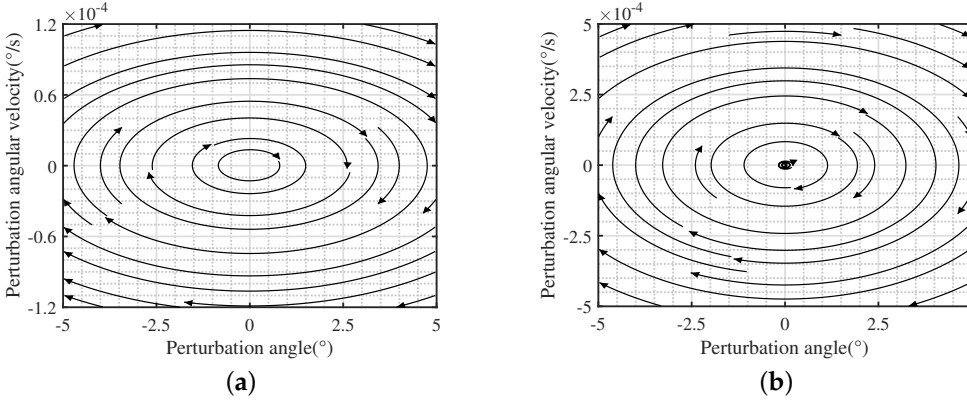

**Figure 9.** Phase diagram of system near $x_0$ ($c_1 = 0$, $c_2 = 0$). (**a**) In-plane swing $\theta$; (**b**) Out-plane swing $\phi$.

According to the above results, it can be found that the characteristic roots of the linearized space elevator system have negative real parts due to the existence of damping such as atmospheric damping and rope damping, so the linearized system is strictly stable. And according to Lyapunov linearization stability determination method, the original nonlinear system is asymptotically stable near the equilibrium point, that is, the segmented space elevator system is asymptotically stable near the equilibrium point.

### 4.2. Stability Range Offset Outside the Equatorial Plane of the Earth

It can be seen from Equations (27) and (30) that the influencing parameters affecting the restoring torque $M_{ro}$ and the tensile force of the anchor point $T_o$ are:

- $a_0$, cross-sectional area of anchor point;
- $m_c$, mass of zenith anchor;
- $p_l$, total length of the space elevator system;
- $p_a$, ratio of the cross-sectional area of the second segment to the first segment;
- $p_s$, location of rope segment.

Through parameter analysis, $p_a$ and $p_s$ have little influence on the restoring torque curve and the anchor point tension curve. The total length of the space elevator rope

and the mass of the zenith anchor are parameters representing the external tension of the synchronous track of the space elevator system, which can be converted to each other. Therefore, $a_0$ and $m_c$ are used to analyze the maximum deflection angle of the lateral displacement of the space elevator system. The basic parameters of the model are shown in Table 4.

**Table 4.** System parameters of the segment space elevator system.

| Parameters | Value |
|---|---|
| $a_0$ | $1.00 \times 10^{-2}$ m$^2$ |
| $m_c$ | $3.48 \times 10^9$ kg |
| $\rho$ | $1.30 \times 10^3$ kg m$^{-3}$ |
| $p_l$ | 9.42 |
| $p_a$ | 2.50 |
| $p_s$ | 1.75 |

The relationship of the restoring torque of the space elevator system with the offset angle is shown in Figure 10.

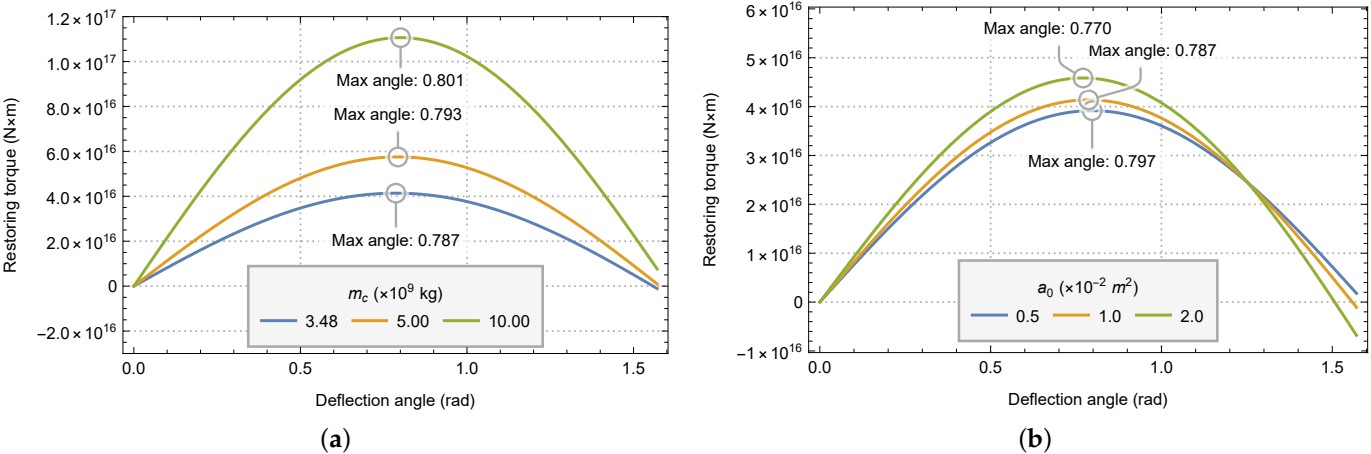

**(a)**                    **(b)**

**Figure 10.** The graph of restoring torque with offset angle under the condition of different parameters. (**a**) With different zenith anchor masses $m_c$. (**b**) With different initial cross-sectional areas $a_0$.

It can be seen from the Figure 10 that the restoring torque of the system first increases and then decreases with the increase of the offset angle. Therefore, when the restoring torque reaches the maximum, the corresponding offset angle is defined as the maximum deflection angle of the system. As can be seen from Figure 10a, with the increase of zenith anchor mass, the value of restoring torque increases, and the maximum deflection angle of the system also increases. From Figure 10b, with the increase of the cross-sectional area of the anchor point, the value of the restoring torque increases slightly, but the maximum deflection angle of the system decreases.

Considering the mass of the zenith anchor and the cross-sectional area of the anchor point, the tension of the anchor point is used as a parameter to analyze the maximum deflection angle outside the equatorial plane of the system. The zenith anchor mass is in the range of $3.50 \times 10^9$ kg to $10.00 \times 10^9$ kg and the cross-sectional area of the anchor point is in the range of 0.001 m$^2$ to 0.02 m$^2$. The anchor point tension of each model is calculated by using the method in [21], and the relationship between the anchor point tension and the system maximum deflection angle is obtained, as shown in Figure 11.

It can be seen from the figure that under the condition of the same anchor point tension, the greater the zenith anchor mass and the smaller the cross section of the anchor point, the greater the maximum deflection angle of the system. Under the same cross-sectional area

of anchor point, the maximum deflection angle of the system and the tension of anchor point will increase with the increase of zenith anchor mass.

By changing the parameters of the horizontal axis in Figure 11 to the stress of the anchor point, Figure 12 can be obtained.

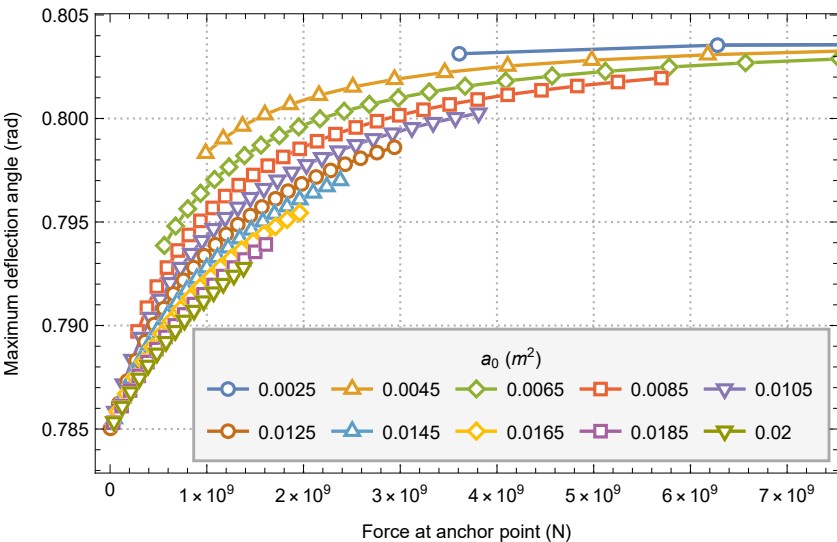

**Figure 11.** Variation diagram of system maximum deflection angle and anchor point tension.

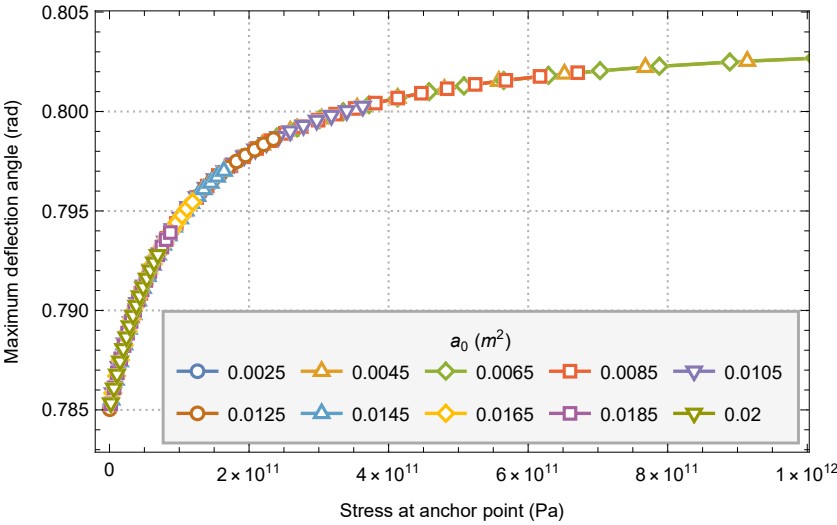

**Figure 12.** Variation diagram of system maximum deflection angle caused by the restoring torque and anchor point stress.

It can be found that the maximum deflection angle of the system is basically the same for the models with different parameters under the same anchor point stress. With the increase of anchor point stress, the maximum deflection angle of the system also increases, but the degree of increase decreases with the increase of anchor point stress. The minimum stable deflection angle of the system is about 0.785 radians, and the maximum deflection angle is about 0.802 radians.

The change of the tensile force at the anchor point of the space elevator system with the offset angle is shown in Figure 13.

With the increase of the offset angle, the tensile force of the anchor point of the system decreases gradually. However, the tensile force of the anchor point of the system will not be reduced to 0, so the offset outside the equatorial plane will not cause the instability of the system caused by the reduction of the tensile force of the anchor point to 0.

When the system is offset out the equatorial plane, the maximum deflection angle is about 0.785 radian to 0.792 radian within a reasonable anchor point stress range (<100 GPa).

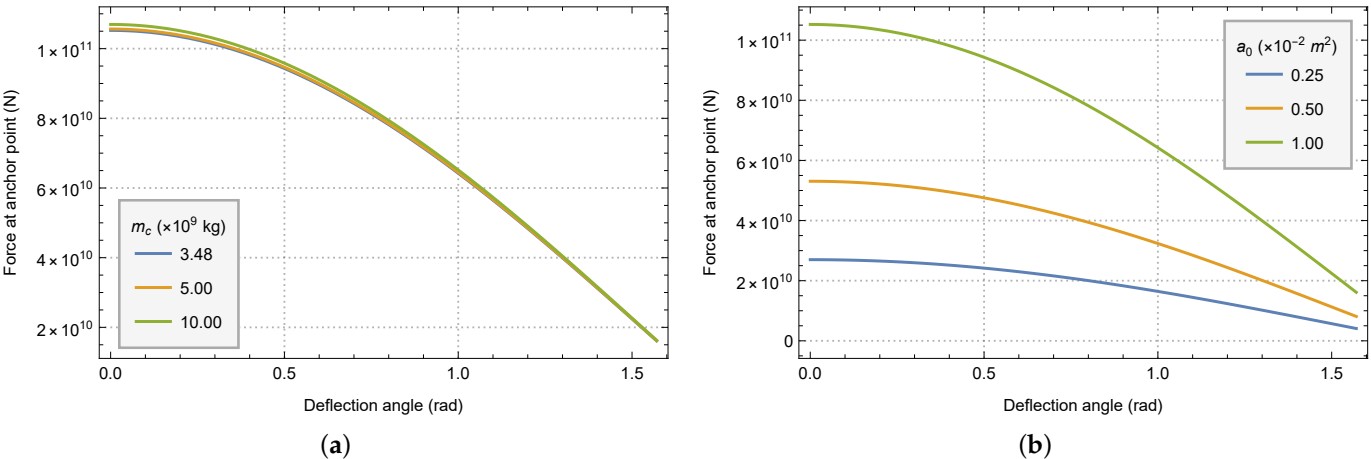

**Figure 13.** The graph of tensile force at the anchor point with offset angle under the condition of different parameters. (**a**) With different zenith anchor masses $m_c$. (**b**) With different initial cross-sectional areas $a_0$.

### 4.3. Stability Range Offset in the Equatorial Plane of the Earth

The same parameters in Table 4 as the out of equatorial offset are used to analyze the offset in the equatorial plane. The relationship of the restoring torque of the space elevator system with the offset angle is shown in Figure 14.

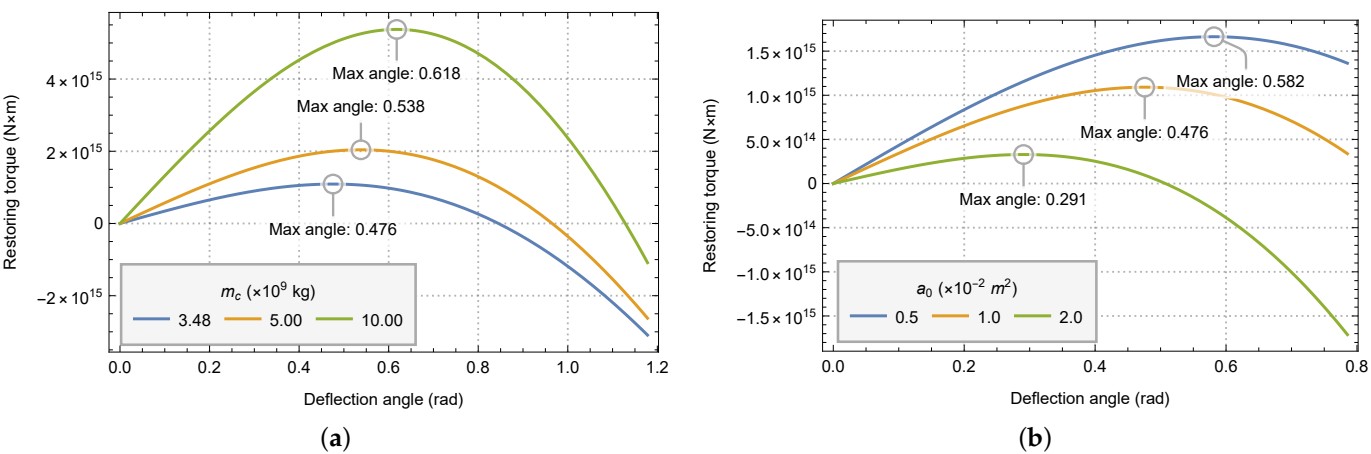

**Figure 14.** The graph of restoring torque with offset angle under the condition of different parameters. (**a**) With different zenith anchor masses $m_c$. (**b**) With different initial cross-sectional areas $a_0$.

It is the same as the case of offset outside the equatorial plane that the restoring torque of the system first increases and then decreases with the increase of the offset angle. With the increase of zenith anchor mass, the value of restoring torque increases, and the maximum deflection angle of the system also increases. With the increase of the cross-sectional area of the anchor point, the value of the restoring torque increases, but the maximum deflection angle of the system decreases. Different from case of offset outside the equatorial plane, the magnitude of the restoring force and the position of the maximum deflection angle vary greatly with the zenith anchor mass and the cross-sectional area of the anchor point.

Similarly, by synthesizing the parameters into the stress of the anchor point, the relationship between the maximum deflection angle and the stress of the anchor point when offset in the equatorial plane can be obtained, as shown in Figure 15.

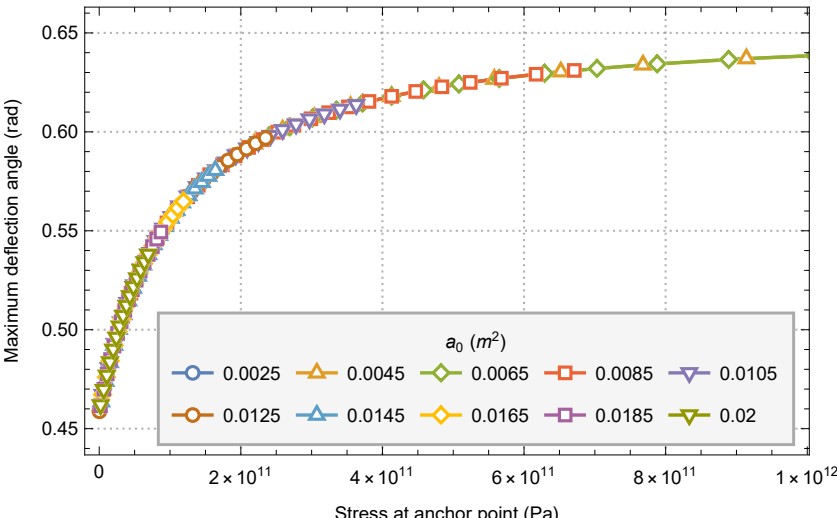

**Figure 15.** Variation diagram of system maximum deflection angle caused by the restoring torque and anchor point stress.

As the case of offset outside the equatorial plane, the maximum deflection angle of the system is basically the same for the models with different parameters under the same anchor point stress. With the increase of anchor point stress, the maximum deflection angle of the system also increases, but the degree of increase decreases with the increase of anchor point stress. However, in the case of offset in the equatorial plane, the minimum maximum deflection angle of the system is about 0.460 radians, and the maximum maximum deflection angle is about 0.640 radians.

The change of the tensile force at the anchor point of the space elevator system with the offset angle is shown in Figure 16.

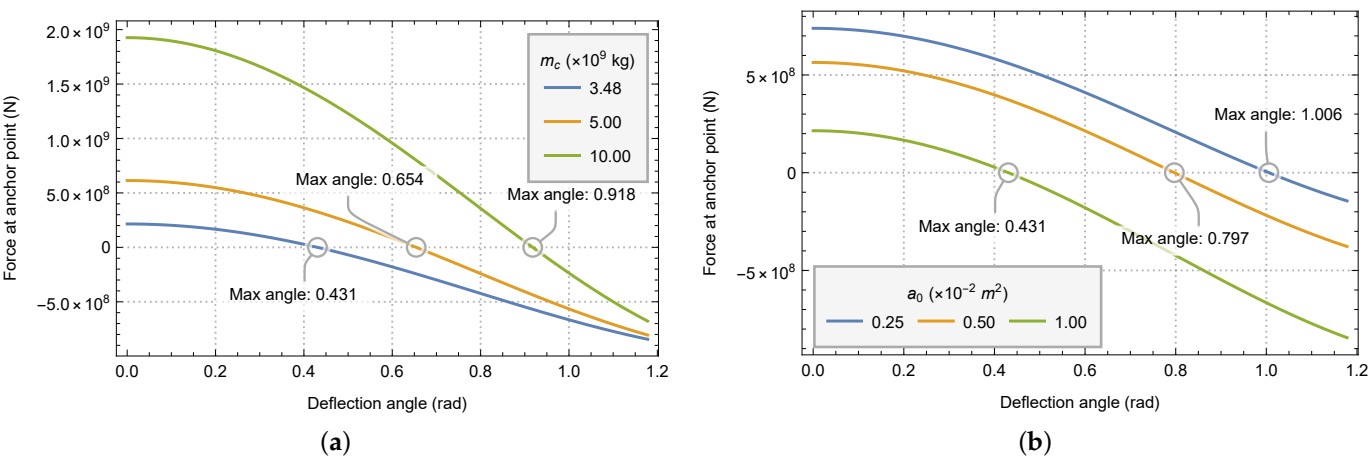

**Figure 16.** The graph of tensile force at the anchor point with offset angle under the condition of different parameters. (**a**) With different zenith anchor masses $m_c$. (**b**) With different initial cross-sectional areas $a_0$.

Different from the offset outside the equatorial plane, with the increase of the offset angle, the tension of the anchor point decreases very rapidly, and becomes 0 after offsetting a certain angle. When the anchor point tension drops to 0, the offset angle at this time is the maximum deflection angle of the system due to the anchor point tension. Similar to the change of restoring torque, the maximum deflection angle caused by the tension of anchor point also changes greatly with the mass of zenith anchor and the cross-sectional area of anchor point.

by synthesizing the parameters into the stress of the anchor point, the relationship between the maximum deflection angle  caused by the tension of anchor point and the stress of the anchor point when offset in the equatorial plane can be obtained, as shown in Figure 17.

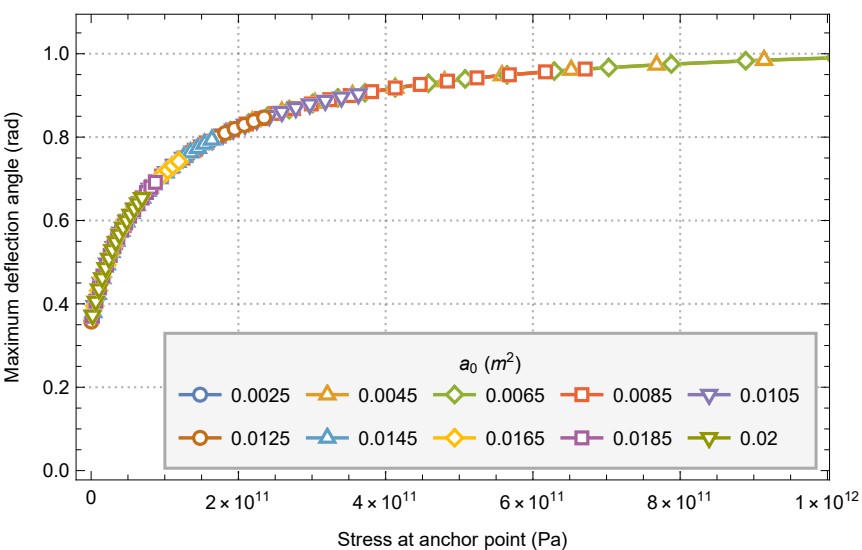

**Figure 17.** Variation diagram of system maximum deflection angle  caused by the tension of anchor point and anchor point stress.

The maximum deflection angle  caused by the tension of anchor point of the system is basically the same for the models with different parameters under the same anchor point stress. With the increase of anchor point stress, the maximum deflection angle  of the system also increases, but the degree of increase decreases with the increase of anchor point stress. The minimum stable deflection angle  of the system is about 0.380 radians, and the maximum deflection angle  is about 0.990 radians.

Combining the maximum deflection angle  of restoring torque and anchor point tension, and limiting the stress of anchor point to a reasonable range (<100 GPa), Figure 18 can be obtained.

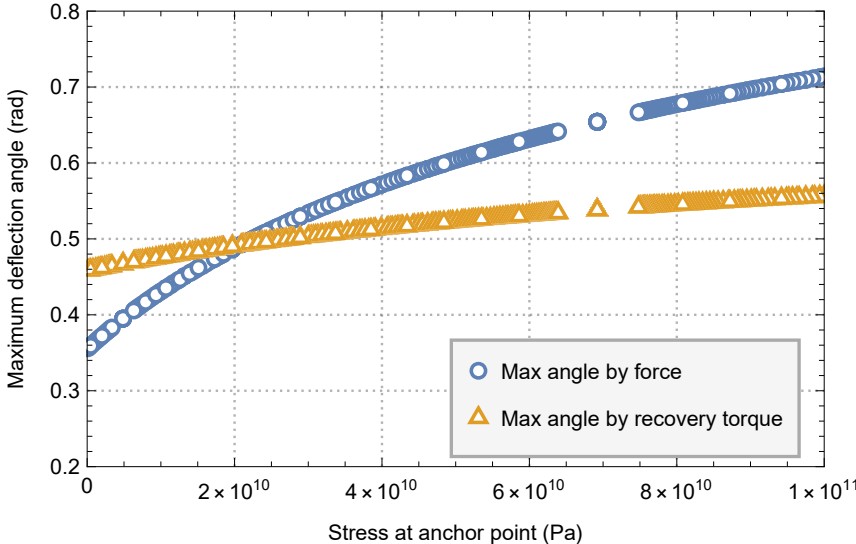

**Figure 18.** Variation diagram of system maximum deflection angle  at the case in the and anchor point stress.

It can be seen from the figure that when the design stress of the anchor point is less than 20 GPa, the main instability of the space elevator system is due to the reduction of the force of the anchor point to 0; When the design stress of the anchor point is greater than 20 GPa, the main instability of the space elevator system is due to the maximum restoring torque. When the system is offset in the equatorial plane, the maximum deflection angle is about 0.36 radian to 0.56 radian within a reasonable anchorage point stress range (<100 GPa).

## 5. Discussion

The rigid rope model is used in this paper. However, when the actual space elevator rope is deformed laterally, it can be seen from [21] that the shape of the rope should be the "s" type. This will lead to the angle deflection of the space elevator rope and a certain horizontal translation, and the decrease of the rope stress will lead to the decrease of the elongation of the rope, which will lead to the increase of the overall gravitation and the decrease of the centrifugal force. This effect has little influence on the restoring torque in the stability of the space elevator system, but has a great influence on the tension of the anchor point. Therefore, the next work is applying the more accurate ANCF rope model to analyze the stability of the space elevator system.

## 6. Conclusions

The segment space elevator system model has the characteristics of easier construction, more practical functions and easier maintenance. Based on the model, in this paper, the stability of the system at the equilibrium point is analyzed by Lyapunov stability theory; And with the criterion that the change rate of the system restoring torque and the anchor point tension are greater than 0, the maximum offset angle of the system inside and outside the equatorial plane is analyzed.

The main conclusions are as follows:

- The segment space elevator is stable near the equilibrium point.
- The maximum deflection angle of the space elevator inside and outside the equatorial plane is related to the design stress of the anchor point.
- When the space elevator is offset outside the equatorial plane, it will only lose stability because the restoring torque reaches the maximum value.
- When the space elevator is offset in the equatorial plane, when the design stress of the anchor point is small, it will lose stability because the tensile force of the anchor point is reduced to 0, and when the design stress of the anchor point is large, it will lose stability because the recovery torque reaches the maximum value.
- The stability of the space elevator outside the equatorial plane is better than that in the equatorial plane.

Stability analysis is a part of the dynamic analysis basis of the space elevator system. This research provides a basis for the dynamic analysis and safety research of the space elevator system, and provides a basis for the future research, such as oscillation suppression, structural design and so on.

**Author Contributions:** Conceptualization, S.L.; methodology, S.L.; software, S.L.; validation, S.L.; formal analysis, S.L.; investigation, Y.F.; resources, Y.F. and N.C.; data curation, S.L.; writing—original draft preparation, S.L.; writing—review and editing, Y.F. and H.G.; visualization, S.L.; supervision, Y.F. and N.C.; project administration, Y.F.; funding acquisition, N.C. and X.W. All authors have read and agreed to the published version of the manuscript.

**Funding:** This research was funded by the National Natural Science Foundation of China (10772057).

**Institutional Review Board Statement:** Not applicable.

**Informed Consent Statement:** Not applicable.

**Data Availability Statement:** Not applicable.

**Conflicts of Interest:** The authors declare no conflict of interest.

## Appendix A

*Appendix A.1. Detail Equations of Offset Outside the Equatorial Plane of the Earth*

The gravity and centrifugal force on the rope of space elevator system can be obtained by integrating Equation (23):

$$
\begin{aligned}
M_{go} &= \int_0^{p_l * R_e} \mathrm{d}M_{go} \\
&= a_0 \mu \rho \csc \phi \left( 1 - \frac{p_a p_l \cos \phi + p_a}{N_{lo}} + \frac{(p_a - 1)(p_s \cos \phi + 1)}{N_{so}} \right),
\end{aligned}
\tag{A1a}
$$

$$
\begin{aligned}
M_{co} &= \int_0^{p_l * R_e} \mathrm{d}M_{co} \\
&= \frac{\mu \rho a_0 R_e^3 \sin \phi}{6 R_g^3} \left( 2 \cos \phi \left( p_a \left( p_l^3 - p_s^3 \right) + p_s^3 \right) + 3 p_a p_l^2 - 3(p_a - 1) p_s^2 \right),
\end{aligned}
\tag{A1b}
$$

where:

$$
N_{lo} = \sqrt{p_l^2 + 1 + 2 p_l \cos \phi},
\tag{A2a}
$$

$$
N_{so} = \sqrt{p_s^2 + 1 + 2 p_s \cos \phi}.
\tag{A2b}
$$

Combining Equations (25) and (A1), the simplified recovery torque $M_{ro}$ can be expressed as

$$
\begin{aligned}
M_{ro} &= M_{go} + M_{tgo} - M_{co} - M_{tco} \\
&= \frac{\mu \rho a_0 R_e^3 \sin \phi}{6 R_g^3} \left( 2 \cos \phi \left( p_a p_l^3 - p_a p_s^3 + p_s^3 \right) + 3 p_a p_l^2 - 3(p_a - 1) p_s^2 \right) \\
&\quad + a_0 \mu \rho \csc \phi \left( \frac{p_a p_l \cos \phi + p_a}{N_{lo}} - \frac{(p_a - 1)(p_s \cos \phi + 1)}{N_{so}} - 1 \right) \\
&\quad + \frac{p_l m_c \mu \sin \phi}{R_e^2} \left( (p_l \cos \phi + 1) \frac{R_e^4}{R_g^3} - \frac{R_e}{N_{lo}^3} \right).
\end{aligned}
\tag{A3}
$$

Similarly, for the force in $X$ direction, overall force can be obtained by integrating Equation (28):

$$
\begin{aligned}
F_{go} &= \int_0^{p_l * R_e} \mathrm{d}F_{go} \\
&= \frac{a_0 \mu \rho}{R_e} \left( \frac{p_a p_l}{N_{lo}} - \frac{(p_a - 1) p_s}{N_{so}} \right),
\end{aligned}
\tag{A4a}
$$

$$
\begin{aligned}
F_{co} &= \int_0^{p_l * R_e} \mathrm{d}F_{co} \\
&= \frac{a_0 \mu \rho}{2 R_e} \left( \cos \phi \left( p_a (p_l^2 - p_s^2) + p_s^2 \right) + 2 (p_a (p_l - p_s) + p_s) \right).
\end{aligned}
\tag{A4b}
$$

Combining Equations (29) and (A4), the simplified force in the $X$ direction $T_o$ can be expressed as

$$
\begin{aligned}
T_o &= F_{co} + F_{tco} - F_{go} - F_{tgo} \\
&= \frac{\mu m_c}{R_e^2}(p_l \cos\phi + 1)\left(\frac{R_e^3}{R_g^3} - \frac{1}{N_{lo}^3}\right) \\
&\quad + \frac{a_0 \mu \rho \cos\phi}{2R_e}\left(p_a(p_l - p_s)(p_l + p_s) + p_s^2\right) \\
&\quad + \frac{a_0 \mu \rho}{R_e N_{so}}\left((p_a p_l - p_a p_s + p_s)\cos\phi + (p_a - 1)p_s\right) \\
&\quad - \frac{a_0 \mu \rho}{R_e N_{lo}}p_a p_l.
\end{aligned}
\tag{A5}
$$

*Appendix A.2. Detail Equations of Offset in the Equatorial Plane of the Earth*

The gravity and centrifugal force on the rope of space elevator system can be obtained by integrating Equation (32):

$$
\begin{aligned}
M_{gi} &= \int_0^{p_l * R_e} \mathrm{d}M_{gi} \\
&= \mu \rho a_0 \csc\theta \cos\theta \left[\frac{(p_a - 1)p_s}{N_{si}} - \frac{p_a p_l}{N_{li}}\right] \\
&\quad + \mu \rho a_0 \csc\theta \left[\frac{(p_a - 1)}{N_{si}} - \frac{p_a}{N_{li}} + 1\right],
\end{aligned}
\tag{A6a}
$$

$$
\begin{aligned}
M_{ci} &= \int_0^{p_l * R_e} \mathrm{d}M_{ci} \\
&= \frac{a_0 \mu \rho R_e^3 \sin\theta \cos\theta}{2R_g^3}(p_a p_l N_{li} - (p_a - 1)p_s N_{si}) \\
&\quad - \frac{a_0 \mu \rho R_e^3 \sin\theta}{4R_g^3}(3\cos 2\theta - 1)(p_a N_{li} - (p_a - 1)N_{si} - 1) \\
&\quad - \frac{3a_0 \mu \rho R_e^3}{2R_g^3}p_a \sin^3\theta \cos\theta \tanh^{-1}\left(\frac{p_l + \cos\theta}{N_{li}}\right) \\
&\quad + \frac{3a_0 \mu \rho R_e^3}{2R_g^3}(p_a - 1)\sin^3\theta \cos\theta \tanh^{-1}\left(\frac{p_s + \cos\theta}{N_{si}}\right) \\
&\quad + \frac{3a_0 \mu \rho R_e^3}{4R_g^3}\sin^3\theta \cos\theta\left[\log\left(\cos^2\frac{\theta}{2}\right) - \log\left(\sin^2\frac{\theta}{2}\right)\right],
\end{aligned}
\tag{A6b}
$$

where:

$$
N_{li} = \sqrt{p_l^2 + 1 + 2p_l \cos\theta},
\tag{A7a}
$$

$$
N_{si} = \sqrt{p_s^2 + 1 + 2p_s \cos\theta}.
\tag{A7b}
$$

Combining Equations (33) and (A6), the simplified recovery torque $M_{ri}$ can be expressed as

$$
\begin{aligned}
M_{ri} &= M_{gi} + M_{tgi} - M_{ci} - M_{tci} \\
&= \frac{\mu p_l m_c \sin\theta}{R_e R_g^3 N_{li}^3} \left( R_e^3 \left( p_l^2 \cos 2\theta + \left( p_l^2 + 3 \right) p_l \cos\theta + 2p_l^2 + 1 \right) - R_g^3 \right) \\
&\quad + \frac{1}{4} a_0 \mu \rho \sin\theta \left( \frac{p_a p_l \cos\theta + p_a}{N_{li}} - \frac{(p_a - 1)(p_s \cos\theta + 1)}{N_{si}} - 1 \right) \\
&\quad + \frac{a_0 \mu \rho p_a N_{li} \sin\theta}{4 R_g^3} (2p_l \cos\theta - 3\cos 2\theta + 1) \\
&\quad + \frac{a_0 \mu \rho (p_a - 1) N_{si} \sin\theta}{4 R_g^3} (-2p_s \cos\theta + 3\cos 2\theta - 1) \\
&\quad + \frac{a_0 \mu \rho R_e^3}{4 R_g^3} \sin\theta (3\cos 2\theta - 1) \\
&\quad + \frac{a_0 \mu \rho R_e^3 \sin\theta}{4 R_g^3} \left[ \log\left( \sin^2 \frac{\theta}{2} \right) - \log\left( \cos^2 \frac{\theta}{2} \right) \right] \\
&\quad + \frac{3 a_0 \mu \rho R_e^3 \sin^3 \theta \cos\theta}{2 R_g^3} \left[ p_a \tanh^{-1} \left( \frac{p_l + \cos\theta}{N_{li}} \right) - (p_a - 1) \tanh^{-1} \left( \frac{p_s + \cos\theta}{N_{si}} \right) \right].
\end{aligned}
\tag{A8}
$$

Similarly, for the force in *X* direction, overall force can be obtained by integrating Equation (28):

$$
\begin{aligned}
F_{gi} &= \int_0^{p_l * R_e} \mathrm{d}F_{gi} \\
&= \frac{a_0 \mu \rho}{R_e} \left[ \frac{p_a p_l}{N_{li}} - \frac{(p_a - 1) p_s}{N_{si}} \right],
\end{aligned}
\tag{A9a}
$$

$$
\begin{aligned}
F_{ci} &= \int_0^{p_l * R_e} \mathrm{d}F_{ci} \\
&= \frac{a_0 \mu \rho R_e^2}{8 R_g^3} (3\cos 3\theta - 7\cos\theta) \\
&\quad + \frac{a_0 \mu p_a \rho R_e^2 N_{li} \cos\theta}{4 R_g^3} (2p_l \cos\theta - 3\cos 2\theta + 5) \\
&\quad + \frac{a_0 \mu (p_a - 1) \rho R_e^2 N_{si}}{8 R_g^3} (-2p_s \cos\theta + 3\cos 2\theta - 5) \\
&\quad + \frac{a_0 \mu \rho R_e^2 \sin^2 \theta (3\cos 2\theta - 1)}{8 R_g^3} \left[ 2(p_a - 1) \tanh^{-1} \left( \frac{p_s + \cos\theta}{N_{si}} \right) - 2p_a \tanh^{-1} \left( \frac{p_l + \cos\theta}{N_{li}} \right) \right] \\
&\quad + \frac{a_0 \mu \rho R_e^2 \sin^2 \theta (3\cos 2\theta - 1)}{8 R_g^3} \left[ \log\left( \cos^2 \frac{\theta}{2} \right) - \log\left( \sin^2 \frac{\theta}{2} \right) \right].
\end{aligned}
\tag{A9b}
$$

Combining Equations (37) and (A9), the simplified force in the *X* direction $T_i$ can be expressed as

$$
\begin{aligned}
T_i &= F_{ci} + F_{tci} - F_{gi} - F_{tgi} \\
&= \frac{a_0 \mu \rho R_e}{R_e^2} \left( \frac{(p_a - 1)p_s}{N_{si}} - \frac{p_a p_l}{N_{li}} \right) \\
&\quad + \frac{\mu m_c (p_l \cos\theta + 1)}{R_e^2 N_{li}^3} \left[ \frac{R_e^3}{R_g^3} N_{li}^2 (p_l \cos\theta + 1) - 1 \right] \\
&\quad + \frac{a_0 \mu \rho R_e^2 \cos\theta \, p_a N_{li}}{4 R_g^3} (2 p_l \cos\theta - 3\cos 2\theta + 5) \\
&\quad + \frac{a_0 \mu \rho R_e^2 \cos\theta (p_a - 1) N_{si}}{4 R_g^3} (-2 p_s \cos\theta + 3\cos 2\theta - 5) \\
&\quad + \frac{a_0 \mu \rho R_e^2}{8 R_g^3} (3\cos 3\theta - 7\cos\theta) \\
&\quad + \frac{a_0 \mu \rho R_e^2 \sin^2\theta (3\cos 2\theta - 1)}{8 R_g^3} \left[ \log\left( \cos^2 \frac{\theta}{2} \right) - \log\left( \sin^2 \frac{\theta}{2} \right) \right] \\
&\quad + \frac{a_0 \mu \rho R_e^2 \sin^2\theta (3\cos 2\theta - 1)}{8 R_g^3} \left[ 2(p_a - 1) \tanh^{-1}\left( \frac{p_s + \cos\theta}{N_{si}} \right) - 2 p_a \tanh^{-1}\left( \frac{p_l + \cos\theta}{N_{li}} \right) \right].
\end{aligned} \tag{A10}
$$

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
