# Peer review of "The Stability Analysis of a Tether for a Segmented Space Elevator"

_aerospace, doi:10.3390/aerospace9070376_

Round 1
Reviewer 1 Report
In this paper, the stability of the system of space elevator the equilibrium point is analyzed by Lyapunov stability theory; based on the criterion that the change rate of the system restoring moment and the anchor point tension is more significant than 0, the maximum offset angle of the system inside and outside the equatorial plane is analyzed. The results show that the segment space elevator is stable near the equilibrium point. Furthermore, the stability limit of the space elevator inside and outside the equatorial plane is related to the design stress of the anchor point. The stability of the space elevator outside the equatorial plane is better than that in the equatorial plane.
Author Response
We sincerely thank the reviewer for thoroughly examining our manuscript and providing very helpful comments to guide our revision.
Reviewer 2 Report
This paper analyzes the stability property of a segmented space elevator using the Lyapunov stability theory. The maximum offset angle of the system inside and outside the equatorial plane is presented as the results. The literature review sufficiently covers relevant papers. Overall, the paper performs a solid derivation and the results are well presented through figures. However, there are a few minor issues that need to be addressed before publication. I recommend accepting with minor revision.
1. Moderate English changes required. Paper writing can be improved. For example: 1) pp. 1, lines 11-13, "When the space elevator is offset in the equatorial plane, when the design stress...". This sentence contains multiple clauses. Please consider simplify the sentence; 2) pp. 2, line 71, "The earth’s non-equatorial space elevator built in the low latitude area of non-equatorial can solve this problem". "this problem" here means which problem? 3) pp. 9, after line 199 and Figure 4, "there is only the change of centrifugal force" -> "the only change is the centrifugal force". etc.
2. Please double-check the mathematical derivation to guarantee accuracy. For example, pp. 5, Eq. (3), why the integration of ds is from Re to l? By definition, "s is the position of the rope element on the tether". "s" has no direct relationship with Re.
Reviewer 3 Report
The paper “The stability analysis of a tether for a segmented space elevator” is devoted to investigation of space elevator dynamics. The paper is not accurately written in some parts, the models used in the work should be justified. So, authors should revise the text deeply before the publication.
The scalar product of the radius-vectors in eq. (3) and (4) should be specified using brackets (r,r) of by transpose sign as rTr.
The dissipation model presented in eq. (5) should be somehow justified or commented. It is not clear what is the nature of that dissipation. Is it the aerodynamic drag effect at equator? What is the estimations on its values? Why is there two different coefficients? More comments are required.
It is not accurately written that the system dynamics equation is x_dot=0 in eq. (14). It seems that in the authors mean that in the equilibrium point all the time derivatives should be zero. Also, the authors wrongly call the equilibrium point as singularity, it is not correct.
It is not evident that the J1 and J2 values in eq. (12) satisfy the relations J1<c1^2/4M1^2 and J2<c2^2/4M1^2, that are required for system stability according to eq. (19). The authors should demonstrate this in the text.
More explanations on the torque dM_co in (21b) is required. What is its meaning? How does this formulae obtained?
The simplified space elevator rod-model used in the paper can be not adequate in case of large deviation angles. For the large angles the flexible motion should be considered in the motion model to study the stability area. Some justification comments should be added by the authors.
The phase diagrams presented in Fig. 5-10 are quite classical and not quite demonstrative, at least by such a number. If the authors leave three types phase diagrams, it will be enough for the text, no need to repeat it for each case.
Everywhere in the text the restoring moment should be replaced by restoring torque.
The value of the cross section a0 as 10^-24 m^2 in table 3 is too small and it is obvious a typo.
It is not clear what is the meaning of the stability limit? What do the authors impose by that phrase? More explanation should be added to the text.
Round 2
Reviewer 3 Report
The authors adressed all the reviewer comments and questions. The paper can be accepted for the publication.